 **eLIFE**

# Chromatin signature of widespread monoallelic expression

**Anwesha Nag[1,2†], Virginia Savova[1,2†], Ho-Lim Fung[3], Alexander Miron[1], Guo-Cheng Yuan[4], Kun Zhang[3], Alexander A Gimelbrant[1,2]\***

[1]Department of Cancer Biology and Center for Cancer Systems Biology, Dana-Farber Cancer Institute, Boston, United States; [2]Department of Genetics, Harvard Medical School, Boston, United States; [3]Department of Bioengineering, University of California, San Diego, La Jolla, United States; [4]Department of Biostatistics and Computational Biology, Dana-Farber Cancer Institute, Boston, United States

**Abstract** In mammals, numerous autosomal genes are subject to mitotically stable monoallelic expression (MAE), including genes that play critical roles in a variety of human diseases. Due to challenges posed by the clonal nature of MAE, very little is known about its regulation; in particular, no molecular features have been specifically linked to MAE. In this study, we report an approach that distinguishes MAE genes in human cells with great accuracy: a chromatin signature consisting of chromatin marks associated with active transcription (H3K36me3) and silencing (H3K27me3) simultaneously occurring in the gene body. The MAE signature is present in ~20% of ubiquitously expressed genes and over 30% of tissue-specific genes across cell types. Notably, it is enriched among key developmental genes that have bivalent chromatin structure in pluripotent cells. Our results open a new approach to the study of MAE that is independent of polymorphisms, and suggest that MAE is linked to cell differentiation.

**\*For correspondence:**
alexander_gimelbrant@dfci.
harvard.edu

†These authors contributed
equally to this work

**Competing interests:** The
authors declare that no
competing interests exist.

**Reviewing editor:** Thomas
Gingeras, Cold Spring Harbor
Laboratory, United States

## Introduction

A variety of genetic and epigenetic factors affect the relative expression levels of the two copies of each given gene in diploid cells. In addition to cis- and trans-regulatory variation (*Gilad et al., 2008*), there are at least three major kinds of non-Mendelian phenomena that control allele-specific expression in mammals. One is the X chromosome inactivation (*Lyon, 1961*): in female embryos, around the time of implantation, about half of the cells choose to inactivate the maternal X, and the rest inactivate the paternal X, affecting most of the X-linked genes (*Carrel and Willard, 2005*; *Berletch et al., 2010*; *Yang et al., 2010*). Another is imprinting: genes such as *IGF2* and *H19* are expressed either from one allele, either paternal or maternal (*Glaser et al., 2006*).

Finally, a significant fraction of mammalian autosomal genes are subject to monoallelic expression (MAE), which reflects a mitotically stable allele-specific expression with different allelic states in clonal lineages. MAE is observed in olfactory receptor genes (*Chess et al., 1994*), as well as genes coding for immunoglobulins and some cytokines (*Pernis et al., 1965*; *Bix and Locksley, 1998*; *Holländer et al., 1998*). Using genome-wide analyses of allele-specific expression, we and others have added a surprisingly large number of the autosomal genes in human and mouse to the MAE class (*Gimelbrant et al., 2007*; *Jeffries et al., 2012*; *Zwemer et al., 2012*; *Li et al., 2012b*), including genes implicated in a number of human diseases, such as Alzheimer's disease (*APP*) (*Bertram and Tanzi, 2012*) and cancer (*DAPK1*) (*Raval et al., 2007*). MAE affects about 10% of ~4000 tested genes in human lymphoblastoid cells (LCLs) and about 15% of more than 1300 assessed genes in analogous mouse cells (*Gimelbrant et al., 2007*; *Zwemer et al., 2012*).

Our growing appreciation of the prevalence of MAE only underscores how little we know about its biology. The only existing, large-scale sets of data are collected in clonal lymphocyte cell lines in vitro.

**eLife digest** Understanding how genes are activated and silenced is one of the central challenges in modern biology. These processes underpin the development of a fertilized egg into a complex organism, and they can also lead to life-threatening diseases when they go wrong. There are two copies of each gene in a human cell, a maternal copy and a paternal copy, and it is thought that both copies are usually regulated together. However, there are exceptions to this rule: for certain genes only the maternal copy is expressed as a protein in some cells, whereas the paternal copy is expressed in other cells.

This form of gene regulation, which is called monoallelic expression, can result in neighboring cells heading down very different paths. In extreme cases, depending on the differences between the two copies of the gene, cells that express one copy may function normally, while cells where the other copy is activated will start forming tumors. However, despite these potentially grave consequences, and early results which suggested that monoallelic expression affected a large number of human and mouse genes, it has proved to be a major technical challenge to identify these genes in most cell types.

Now, Nag, Savova et al. have discovered a molecular signature that can be used to detect monoallelic expression. The signature was found in chromatin, the densely packed structure formed by DNA and proteins inside the cell nucleus. Nag, Savova et al. discovered that the genes that are subject to monoallelic expression are bound with proteins that are modified in two contrasting ways. One modification, which is usually a sign of gene silencing, is prevalent on the inactive copy of the gene, and the other, which often marks active genes, is chiefly present on the active copy.

Nag, Savova et al. report that these modifications are found in different sets of genes in different cell types, indicating distinct genome-wide patterns of monoallelic expression. The chromatin signature approach lets them estimate the fraction of human genes that are subject to monoallelic expression. This number is surprisingly high: about 20% of commonly expressed genes and more than one-third of tissue-specific genes. In a particularly intriguing finding, almost all bivalent genes—a subset of genes that are involved in determining the fate of cell during development—are estimated to become monoallelic when they are activated.

In addition to these unexpected findings, the chromatin signature approach opens the door to exploring monoallelic expression as a form of gene regulation in all types of cells and, ultimately, to understanding how it is involved in both normal development and in disease.

The limited number of analyzed clones is insufficient to generate a complete catalog of MAE genes in that cell type, and little is known about the prevalence of MAE in other cell types. Virtually nothing is understood about the establishment of MAE during development and differentiation. Mechanistically, allelic choice has been linked to changes in chromatin states in some special cases: imprinting (*Wen et al., 2008*), olfactory receptor gene choice (*Magklara et al., 2011*), and immunoglobulin-kappa gene rearrangement (*Farago et al., 2012*). In contrast, for hundreds of other autosomal MAE genes, no molecular features have been associated with establishment and maintenance of allelic choice. Similarly, there is no general understanding of MAE's function.

A major technical bottleneck in addressing these questions is the clonal nature of MAE (*Figure1—figure supplement 1*). Like X inactivation, MAE is masked in polyclonal samples, and obtaining monoclonal cell populations is challenging for most tissue types, particularly so in vivo. Moreover, genome-wide methods are limited by the availability of polymorphisms. In this study, we report a fundamentally new approach to the detection of monoallelic expression. In contrast to other methods, it does not require any allele-specific information, instead relying on a specific chromatin pattern as a proxy for MAE. We use this approach to address questions about MAE's prevalence, development, and function.

## Results

### MAE genes have a characteristic chromatin signature

Histone modifications, in their diversity, present rich combinatorial possibilities for controlling gene transcription (*Barski et al., 2007*; *Ernst and Kellis, 2010*; *Filion et al., 2010*). They therefore offer

a constrained yet rich set of data for systematic analysis. To identify histone modifications that might be specific to the MAE genes, we compared chromatin marks associated with known MAE genes against those for known biallelically expressed (BAE) genes.

Previous observations in human and mouse cells suggested that a gene could show MAE in one cell type but not another (*Gimelbrant et al., 2005*; *Gimelbrant and Chess, 2006*; *Gimelbrant et al., 2007*). We therefore hypothesized that if there were a correspondence between MAE state and chromatin modifications, it would be more pronounced when comparing data from the same cell type, while not necessarily from exactly the same cells. Since the largest available sets of known human MAE and BAE genes were identified in human lymphoblasts (*Gimelbrant et al., 2007*) (see also Dataset S1 in Dryad, *Nag et al., 2013*), we used these sets to compare with histone modification data for GM12878 lymphoblastoid cells deposited by the ENCODE project (*Dunham et al., 2012*).

We focused on the eight marks that were investigated in a broad variety of cell types: H3K27me3 (histone H3 Lys-27 trimethylation), H3K36me3, H3K4me2, H4K20me3, H3K27ac (histone H3 Lys-27 acetylation), H3K4me1, H3K4me3, H3K9ac (*Figure 1—figure supplement 2A*). To enable our analysis, we reduced complex patterns of histone modifications to two simple features capturing signal intensity in two distinct spatial domains: the proximal promoter signal (for any given modification: ChIP-Seq signal intensity integrated over the 2.5 kb region upstream of transcription start) and the signal integrated over the whole gene body (green and red areas in *Figure 1A*; see 'Materials and methods', *Figure 1—figure supplement 2*). We then set out to analyze in a systematic way whether some combination of these measured signals can reliably distinguish known MAE genes and known biallelic genes.

Using all available features, we wanted to identify parameters that would offer the optimal trade-off between low false positive rate and the largest possible number of correctly identified MAE genes. After systematic exploration detailed in *Figure 1—figure supplement 2B*, we chose two relative penalty tradeoff settings: 'Neutral' (1:1) and 'Precision' (8:1). The best-performing classifier, Decision Tree (DT), identified at the more relaxed 'Neutral' stringency setting almost 80% of the known MAE genes with 20% false positive rate. At a more stringent, precision-optimized setting, the DT classifier identified only 10% false positives, maintaining ~60% of true positives (*Figure 1—figure supplement 2B*).

Strikingly, performance of the DT classifier with just two gene-body features, H3K27me3 (associated with gene silencing) and H3K36me3 (associated with gene transcription), was as good as the performance of any full-featured model (*Figure 1—figure supplement 2C*). Other feature combinations also had some discriminatory power, but not as significant (*Figure 1—figure supplement 3*). This suggests that H3K27me3 and H3K36me3, taken together, account for most of the distinction between MAE and BAE genes. *Figure 1B* also illustrates the partition of the phase space defined by these two signals by the most optimal classifier (two-feature Decision Tree; DT2F; see 'Materials and methods') at both high and medium stringency settings (Precision and Neutral). Performance of the DT2F classifier was at least as good as the performance of any classifier, including any other tested feature combination (*Figure 1—figure supplement 2*).

We concluded that known autosomal MAE and BAE genes in this study show consistent, characteristic differences with respect to the gene body ChIP-seq signal for H3K27me3 and H3K36me3. Co-occurrence of these two signals in the same gene body is thus a specific chromatin signature of known MAE genes. Importantly, MAE identified in clones is reflected by the chromatin signature in a polyclonal sample.

## Chromatin signature is a reliable predictor of MAE

To broadly test this chromatin signature as a predictor of the MAE state, we identified novel genes with the MAE chromatin pattern, and assessed their allelic bias in monoclonal cell lines. Specifically, we calculated gene-body signal for H3K27me3 and H3K36me3 for all autosomal RefSeq genes based on the ENCODE ChIP-seq data for GM12878 lymphoblastoid cells (*Figure 1C*). In these cells, the classifier predicted as monoallelic 1315 (13%) of 10,322 autosomal genes with moderate or higher expression (RPKM > 1), and 15 genes (3%) of 450 highly expressed genes (with RPKM > 100) (*Figure 1D*; detailed in Dataset S1 in Dryad, *Nag et al., 2013*). As a group, the predicted MAE genes were spread throughout all autosomes. They were interspersed with biallelic genes and showed a variety of expression levels (*Figure 1E*). This is consistent with MAE genes identified earlier (*Gimelbrant et al., 2007*; *Zwemer et al., 2012*).

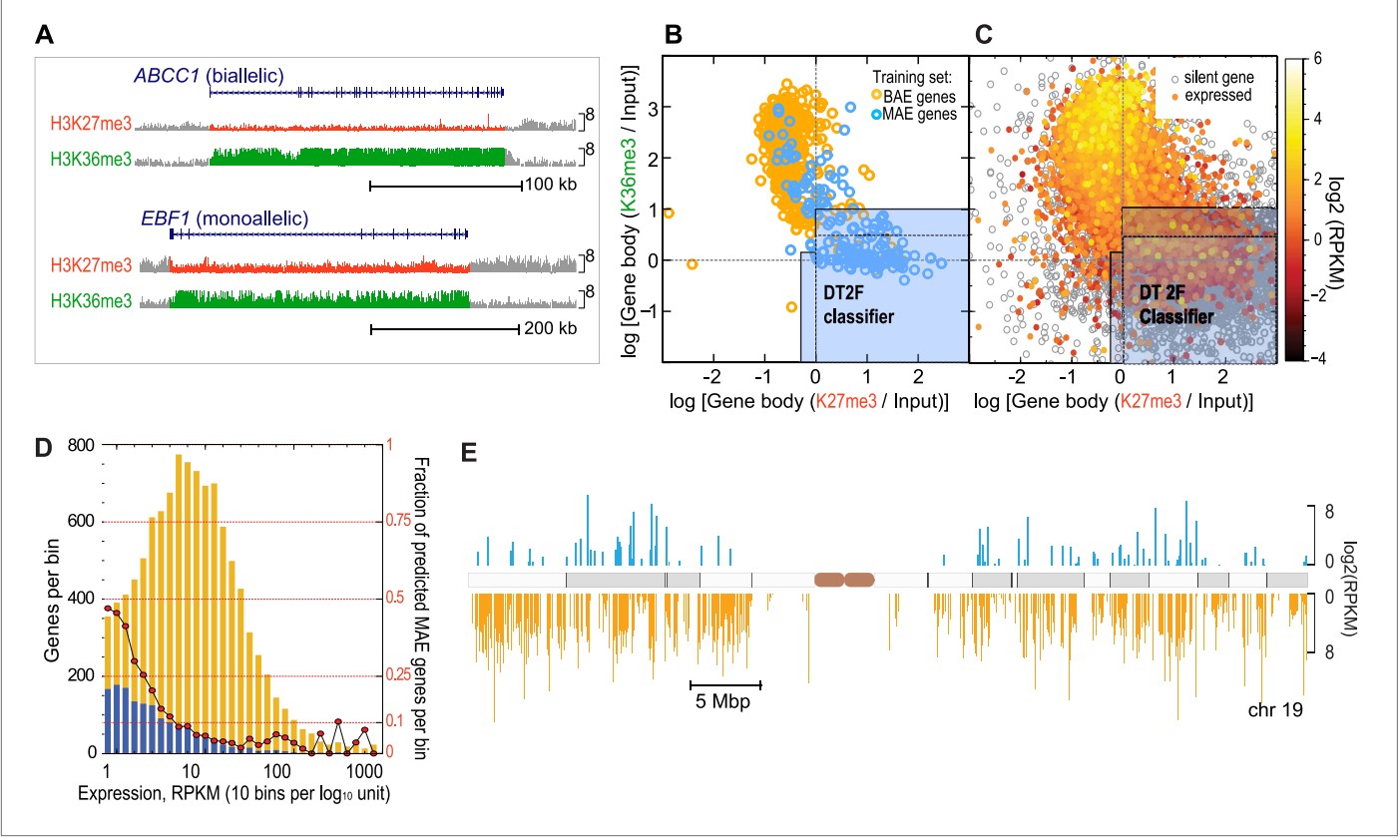

**Figure 1**. Genes with monoallelic expression have specific chromatin signature within the gene body. (**A**) Assessment of histone modifications. The mapped ChIP-Seq signals for the listed modifications were derived from the total signal over the gene-body or promoter region: shown is the gene body signal for the two most informative chromatin marks H3K36me3 (*green*) and H3K27me3 (*red*). *EBF1* gene was shown to be MAE, *ABCC1* was shown to be biallelic in lymphoblastoid cells (*Gimelbrant et al., 2007*). ChIP-Seq data in GM12878 lymphoblasts were generated by the ENCODE project. Graphics adapted from UCSC genome browser (http://genome.ucsc.edu/; *Meyer et al., 2013*). Height of the signal tracks was set 0–8. (**B**) High confidence MAE (blue) and biallelic (gold) autosomal genes in the training set are separated by the gene body signal for H3K27me3 and H3K36me3 in GM12878 cells. Light blue area illustrates partitioning of this space by the most optimal classifier (DT2F). Solid line demarcates external border of 'Neutral' setting; dotted line shows more restrictive 'Precision' setting and is a graphical representation of the boundary identified by an alternating decision tree (DTree), which was the best-performing machine learning method applied to the features after feature selection. Of 270 high confidence MAE genes, 268 had data for both H3K27me3 and H3K36me3. Of these, 204 (76%) are within predicted MAE region. (**C**) Distribution of all autosomal RefSeq genes in GM12878 cells according to gene body signal for H3K27me3 and H3K36me3. Genes are color-mapped according to their expression level in GM12878 cells, from lowly expressed in red to highly expressed in yellow. Silent transcripts (RPKM <= 0.1) are shown in gray. Solid and dotted lines as in 1B. (**D**) Fraction of predicted MAE genes as a function of gene expression level. Left vertical axis: absolute number of predicted MAE (blue) and non-MAE genes (gold) per expression level bin. Right axis: fraction of predicted MAE genes (red circles) per same bin. Expression bins are 0.1 log10 units of RPKM in GM12878 cells. (**E**) Genome distribution of predicted MAE and biallelic genes and their expression level. Shown is chromosome 19; other autosomes are similar. Blue—genes predicted as MAE; gold—genes predicted as biallelic. Position along the chromosome corresponds to transcription start site of the gene; marker length reflects gene expression level in GM12878 cells. Only genes with RPKM > 1 are shown.

The following figure supplements are available for figure 1:

**Figure supplement 1**. Chromatin signature of monoallelic expression allows its detection in monoclonal and polyclonal samples.

**Figure supplement 2**. Building and performance of chromatin feature classifiers.

**Figure supplement 3**. Distribution of various promoter and/or normalized gene body signal combinations in GM12878 cells in our training set.

To test for transcriptional allelic bias in predicted MAE genes, we used two complementary approaches. For a broad, genome-wide analysis, we used mRNA-seq in two independent (with different direction of X-inactivation) monoclonal cell lines, DF1 and DF2. We derived these lines from GM12878

line (see 'Materials and methods'), because both its own genome and its parental genomes were fully sequenced by the 1000 Genomes Project (*Consortium, 2010*). We generated the RNA-seq data from the DF1 and DF2 clones and analyzed them using a custom analysis pipeline (see 'Materials and methods'). Of the 48 X-linked genes with SNPs covered by 10 or more reads, 43 (~90%) showed positive evidence of clone-specific allelic bias; the rest were inconclusive; positive evidence for equivalence was not detected in any X-linked genes (Dataset S2 in Dryad, *Nag et al., 2013*). By contrast, of the 3270 autosomal genes with comparable coverage, 1167 (35%) showed positive evidence of equivalent expression of both alleles and no evidence of bias in either clone; 269 (8%) had allelic bias, while the rest were inconclusive.

Altogether, there were 5001 autosomal genes with at least one SNP covered at any level (Dataset S2 in Dryad, *Nag et al., 2013*). Of these, 1021 genes were predicted to be MAE by the DT2F classifier at the Neutral setting and 236 genes at the Precision-optimized setting. For a quantitative bias comparison, we used all expressed genes. For additional control, we also used equally sized sets of genes with matched levels of expression, randomly chosen from these remaining genes. Both control groups showed quite small mean allelic bias, about 60:40. By contrast, genes predicted MAE by the Neutral DT2F classifier showed about 75:25 bias, and the genes predicted by the classifier at Precision setting had mean bias of 90:10 (*Figure 2B*). To estimate probability of error, we sampled 10 sets of 40 predicted genes and 10 equally-sized control sets, matched by expression; the comparison of the mean bias in the 10 sampled sets showed highly significant difference (p<9e−05; non-paired *t* test). Thus the predicted MAE genes had significantly higher bias than the control genes. In subsequent analysis, we used the neutral classifier setting in order to maximize the number of candidate MAE genes and scrutinize predictive properties of this less stringent setting.

Next, we used the RNA-Seq data to categorize predicted and control genes as biased, unbiased, or indeterminate (*Figure 2C*). Biased expression was identified based on FDR-corrected binomial testing and allelic skewing of at least 2:1 (see 'Materials and methods'). Importantly, rejection of the bias hypothesis by this test does not automatically mean the gene could be called unbiased. Therefore, we used equivalence testing (*Limentani et al., 2005*), with equivalence boundaries corresponding to the two-fold imbalance; genes that failed both tests were called indeterminate. Genes predicted by the DT2F neutral classifier were enriched for genes with positively identified allelic bias; the precision classifier setting, as expected, yielded still better enrichment but fewer positively identified genes (*Figure 2D,E*).

This RNA-Seq approach confirms MAE predictions on a whole-transcriptome level, but it has significant limitations. Insufficient coverage depth leaves an overwhelming majority of genes as 'indeterminate' (*Figure 2E*). This results in underestimation of both the true positive and the true negative rates. Furthermore, a large majority of known MAE genes (about 85%) show biallelic expression in some clonal lineages (*Gimelbrant et al., 2007*; *Zwemer et al., 2012*). This is highly important when considering any validation experiments, since even exhaustive analysis of just two independent clones would miss monoallelic expression in many such genes that would happen to show biallelic expression in the two assessed clones. To validate MAE predictions more conclusively, we measured allelic bias in a greater number of independent clones. To simultaneously increase both coverage depth and the number of biological samples, we designed a targeted extra-deep RNA-Seq assay (allele-specific targeted sequencing; AST-Seq) that allowed us to precisely quantify allele-specific expression of a subset of genes in an increased number of clones (see *Figure 3A*).

To assess both false negative and false positive rates for predictions by the DT2F classifier, we chose a set of predicted, unconfirmed MAE genes expressed in both assessed clones, and a comparable random set of predicted biallelic genes (see 'Method note 2'). Previously, we had derived and characterized several independent clones from GM13130 lymphoblastoid cells (*Gimelbrant et al., 2007*). Starting with four of these clones and the two clones from GM12878, we selected SNPs heterozygous in both genotypes. To control for possible genotyping errors and amplification bias, we used genomic DNA from the same cells. After removing SNPs that did not pass the equivalence test in the gDNA (cf. *Figure 2C*), we had SNPs in 17 predicted MAE genes and 28 predicted biallelic genes. As templates, we used DNA and DNase-treated nuclear RNA from original cell lines and the clones (*Gimelbrant et al., 2007*); as a positive control for expression bias, we included X-linked genes Dataset S3 in Dryad (*Nag et al., 2013*).

We performed two sequencing experiments, with overlapping samples, obtaining 1.6 M reads on average per sample per run. *Figure 3B* shows a representative subset of all classes of assessed genes and associated allelic counts (complete data in Dataset S3 in Dryad, *Nag et al., 2013*). While alleles were equally represented in genomic DNA, the X-linked genes (e.g., *PIR* and *XIAP*) showed strong

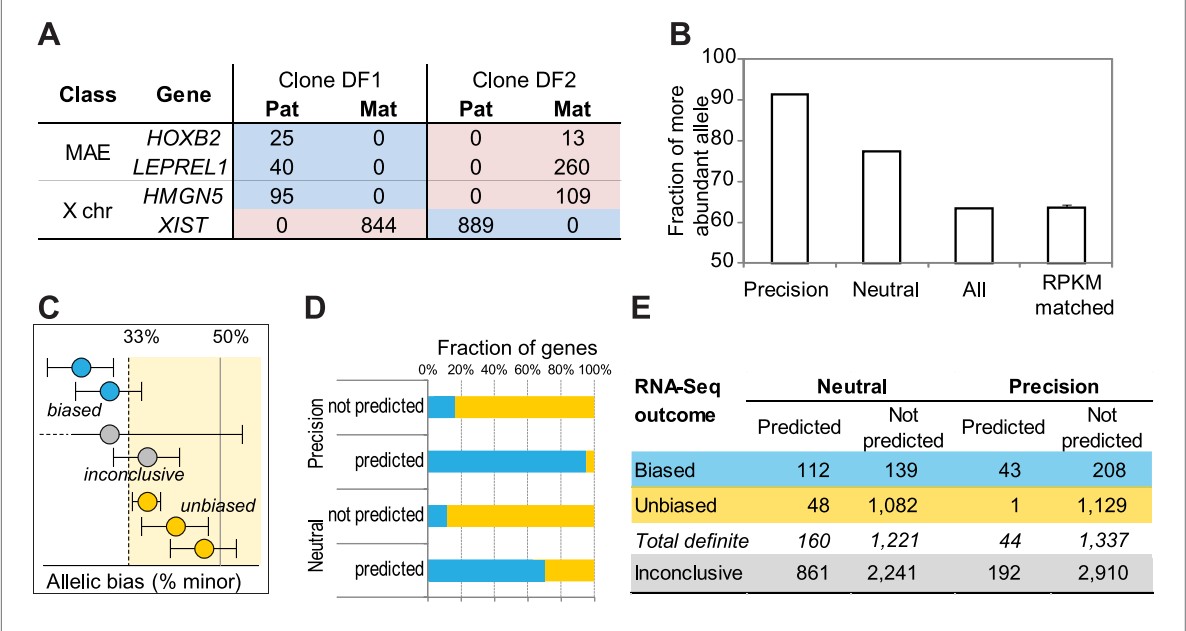

**Figure 2**. Prediction testing with RNA-Seq. (**A**) Representative examples of allelic counts in data from two clones (DF1 and DF2) derived from GM12878 cells. Shown are total maternal (Mat; 'pink') or paternal (Pat; 'blue') counts for X-linked genes and autosomal monoallelic genes illustrating that the direction of allelic bias is clone-specific. (**B**) Mean allelic bias in different groups of genes in DF1 and DF2 clones as assessed by the RNA-Seq analysis. '50' corresponds to perfect balance between alleles; '100' to perfectly monoallelic expression. 'Precision' and 'Neutral'—all informative expressed genes predicted as MAE using corresponding settings of the DT2F classifier; 'All'—all informative expressed genes from GM12878 cells; 'RPKM matched'— predicted biallelic genes, matched by the expression level to the predicted MAE genes (shown are the mean and standard deviation for 10 permuted sets of genes). (**C**) Definitions of allelic bias and lack of bias. Unbiased genes (gold) pass equivalence test (with 2:1 boundaries in either direction; equivalence area is light yellow); biased (blue) pass binomial test; genes that pass neither statistical test are called inconclusive (gray). See **Figure 2—figure supplement 1** for X-chromosome analysis results according to this scheme. (**D**) Fraction of genes showing allelic bias in DF1 and DF2 clones as assessed by RNA-seq. Biased genes were called in at least one clone based on FDR-corrected binomial test and displayed at least 2:1 bias. Unbiased genes were called based on passing the equivalence test in at least on clone and not passing the bias test in the other clone. (**E**) Allele-specific analysis of RNA-Seq data from DF1 and DF2 clones. Experimentally determined allelic states of autosomal genes. Predicted monoallelic and biallelic status is based on the neutral DT2F classifier. Assignments of genes as biased, unbiased or inconclusive (indeterminate in both clones). Color-coding as in panel **C**.

The following figure supplements are available for figure 2:

**Figure supplement 1**. Allelic bias calling on X-Chromosome.

clone-specific bias; genes predicted as biallelic (e.g., *CEP110* and *HDLBP*) showed no significant bias. Most autosomal genes predicted as monoallelic (e.g., *FRMD6* and *IGF1R*) showed the characteristic pattern of clone-specific monoallelic expression: strong bias in some clones, and biallelic expression or opposite bias in others.

The complete results of this experiment (see Dataset S3 in Dryad, **Nag et al., 2013**) are summarized in **Figure 3C**. Of the 17 predicted autosomal MAE genes, 13 exhibited strong allelic bias in at least some of the assessed clones, with 12 of those showing different direction of bias between individual clones (false positive rate of about 24%). Note that the false positive rate should be treated as an overestimate, since two of the unconfirmed genes failed in the GM13030-derived clones, while two others failed altogether. In addition, the number of clones we assayed, while higher than in the first experiment, is not sufficient to establish at high level of certainty that these are truly biallelic genes. Conversely, of the 28 predicted biallelic genes, none showed significant allelic bias in expression in any of the clones. We can thus conclude that the MAE chromatin signature is a specific and sensitive predictor of clone-specific MAE status.

## Asymmetric distribution of chromatin marks between the active and inactive alleles

We asked how the allelic distribution of the active and repressive chromatin marks in clonal cell lines relates to the transcriptional allelic bias. A multiplexed padlock probe approach (**Zhang et al., 2009**)

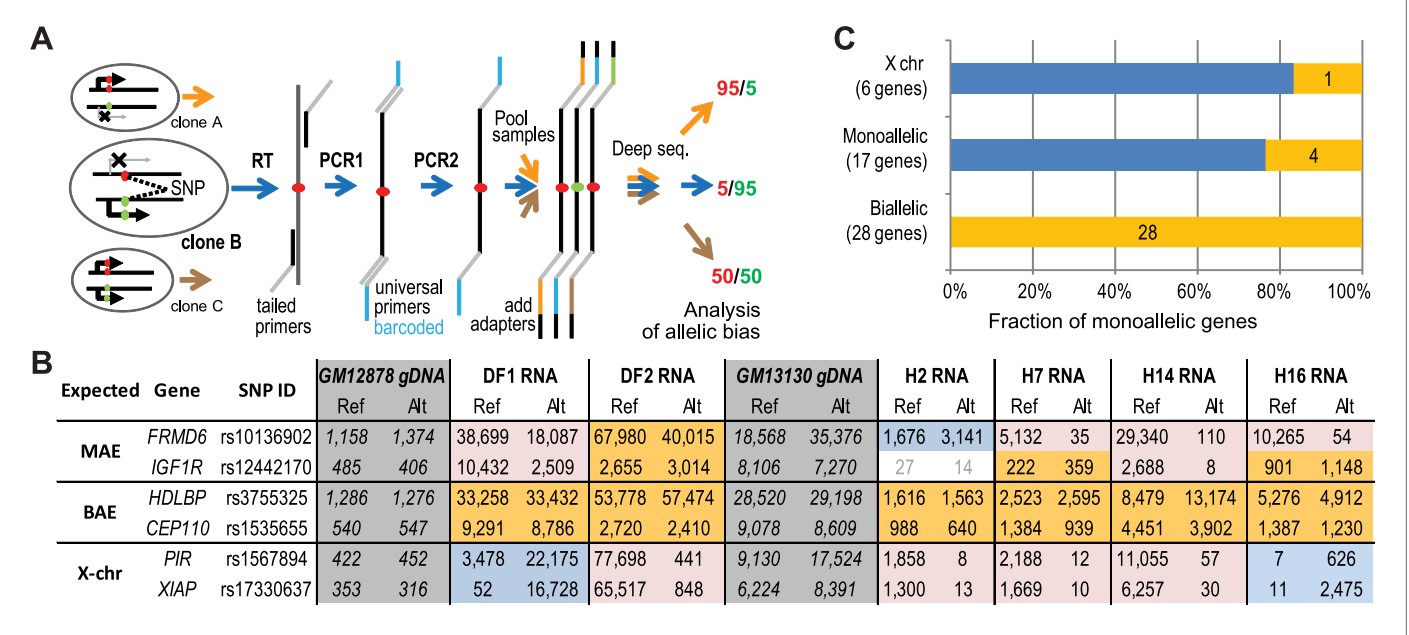

**Figure 3**. Prediction testing with allele-specific targeted sequencing (AST-Seq). (**A**) Schematic representation of the deep barcoding approach. As an illustration, analysis of three clones with no multiplexing is shown, each with a different allelic bias at a SNP of interest. Random-primed cDNA or genomic DNA are used as templates for PCR1, using gene-specific primers with universal tails. The next step associated universal amplicon tails in each sample with two barcodes (PCR2); this allows for barcoding a large number of samples with limited number of secondary primers. For a given sample, all amplicons share the same two barcodes. Barcoded amplicons from all samples are pooled, and sequencing adaptors attached. After sequencing and deconvolving by barcode, allelic hits are counted. (**B**) Representative allelic counts using the AST-seq. Allelic bias was assessed in two clonal lines, DF1 and DF2, derived from GM12878 and four clones, H2, H7, H14, and H16, from GM13130 ('H0') cells. Target SNPs were chosen to be informative in both cell lines. Genomic DNA (gray) was used as a control for allelic bias introduced in amplification; only unbiased assays were pursued. Shown are representative assays for X-linked genes (as control), and examples of genes predicted MAE or biallelic based on the chromatin signature in GM12878. Pink: expression bias towards reference (Ref) allele; blue: expression bias towards alternative (Alt) allele; gold: unbiased expression; no color: counts below threshold—data ignored. Note that, as expected, genes with clone-specific MAE could be biallelic in some clones. (**C**) Summary of the AST-seq analysis for all tested genes in six clonal samples. Biased (blue) and unbiased (gold) expression as defined in *Figure 2C*.

enabled us to assess allelic bias in heterozygous exonic SNPs in two clones with GM12878 genotype, and four clones from GM13130 cells. After removing assays that failed equivalence test in gDNA in all samples, we had 482 SNPs (see Dataset S4 in Dryad, *Nag et al., 2013*). We used this approach to assess allelic bias in H3K27me3 and H3K36me3 ChIP samples simultaneously with cDNA from the same cells, as well as ChIP input and genomic DNA controls.

*Figure 4A* shows representative examples of allelic counts for all classes of expression bias using a small number of SNPs in the two GM12878 clones (complete dataset in Dataset S4 in Dryad, *Nag et al., 2013*). Allelic expression bias is evident in an imprinted gene *SNRPN*, on the X chromosome (*SLC25A43* and *XIAP*), and on autosomal MAE genes. Biased expression is accompanied by an asymmetric distribution of the two chromatin marks: H3K36me3 is associated with the higher transcribed allele, while H3K27me3 with the fully or partially silenced allele.

We used the complete dataset Dataset S4 in Dryad (*Nag et al., 2013*) to evaluate global relationships between expression bias and chromatin allelic bias. In order to pool data from two individuals, one of which (GM13130) lacked complete genotypes for parents, we assessed SNP bias as reference and alternative alleles (rather than maternal or paternal bias). SNPs in cDNA were assigned to one of three bins: reference allele bias; no bias; and alternative allele bias. For these groups, allelic bias in H3K27me3 (*Figure4B*) and H3K36me3 (*Figure 4C*) was determined. In unbiased loci, both H3K27me3 and H3K36me3 were equally represented. In contrast, preferential expression of an allele was associated with elevated levels of H3K36me3 and decreased levels of H3K27me3 on that allele. Both effects were highly significant (p<2 × 10e−9).

Genes predicted to have MAE were about fourfold over-represented among genes where SNPs showed significant bias (*Figure 4D*). SNPs with skewed H3K27me3 and H3K36me3 distribution were

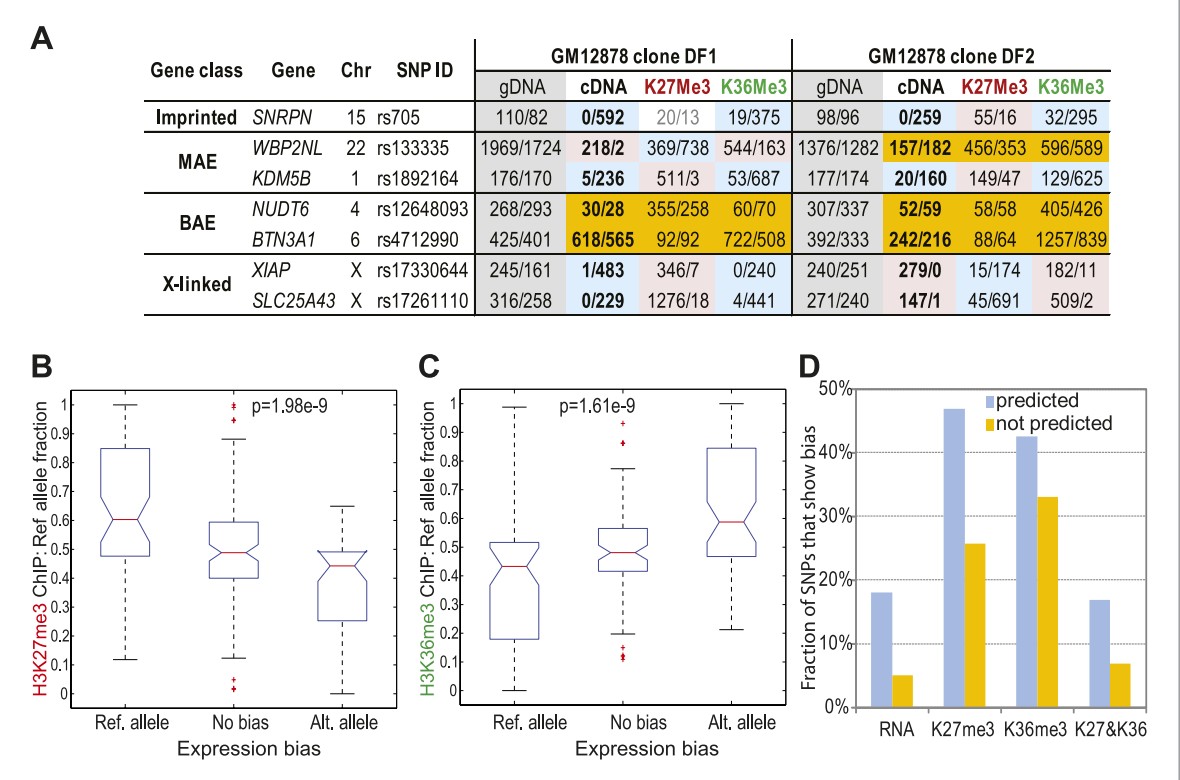

**Figure 4**. Correlation of allelic bias in expression with bias in chromatin marks. (**A**) Representative examples of allelic counts in SNPs assessed with multiplexed targeted sequencing using padlocked probes. Apart from shown clones, additional clones from GM13130 individual were assessed (see 'text'). Shown as control are an imprinted gene *SNRPN* and X-linked genes. Other genes were predicted MAE or biallelic based on their gene-body chromatin signature in GM12878 cells. Measurement for each SNP is shown as read counts for the reference and alternative (Ref/Alt) alleles as designated in dbSNP. Color-coding as in *Figure 3B*. Analysis summarized in this figure is based on 482 SNPs within 458 genes. (**B**) Correlation of allelic bias in H3K27me3 with allelic bias in cDNA. All informative SNPs were put into one of three bins according to their cDNA allelic bias: unbiased (*Figure 2C*); significantly biased towards reference allele; significantly biased towards alternative allele. For each of these groups, allelic bias for the ChIP sample from the same clone was assessed and analyzed using Kruskal–Wallis non-parametric ANOVA test. (**C**) Same as **B**, for H3K36me3. (**D**) Fraction of SNPs in predicted MAE and biallelic genes, showing allelic bias of 2:1. Bias calls were made by binomial testing in cDNA, H3K36me3 and H3K27me3 ChIP for SNPs, using data from the padlock probe experiments.

highly enriched in the genes predicted as MAE (p<10e−6 and p=0.01, respectively; two-tailed Fisher's exact test). This suggests that the asymmetric distribution of the histone modifications is to a large extent due to the genes that have the chromatin signature of monoallelic expression.

## Chromatin signature of MAE shows tissue-specific pattern

Using RNA–DNA FISH, we have previously shown (*Gimelbrant et al., 2007*) that individual MAE genes identified in lymphoblasts also show monoallelic expression in vivo, in peripheral blood mononuclear cells (PBMCs). We used the trained DT2F classifier to analyze available ChIP-seq data from PBMCs (*Bernstein et al., 2010*). In this and the following analyses we applied the classifier to all genes with evidence of transcription (excluding only genes that were not expressed; with RPKM<0.1), to achieve the most comprehensive possible coverage ('Method note 1'). The overall distribution of genes in the DT2F phase space was very similar in cells in vitro and in vivo, with two principal clusters at high H3K36me3 and high H3K27me3 (*Figure 5A*). Predicted genes in PBMC largely overlapped with our predictions for LCLs: out of 2057 genes showing the MAE chromatin signature in PBMC, and called in GM12878 cells, 83% (1712) were also predicted MAE there. This is particularly striking since the experimental data for these two related cell types were collected in two different laboratories. This suggests that the genes with MAE signature in lymphoblasts are quite similar to those in the related ex vivo cells.

We then asked if that similarity is in fact due to biological relatedness of LCLs and PBMCs; alternatively, it could be due to the signature being uniform in all cell types. We applied our analysis to all

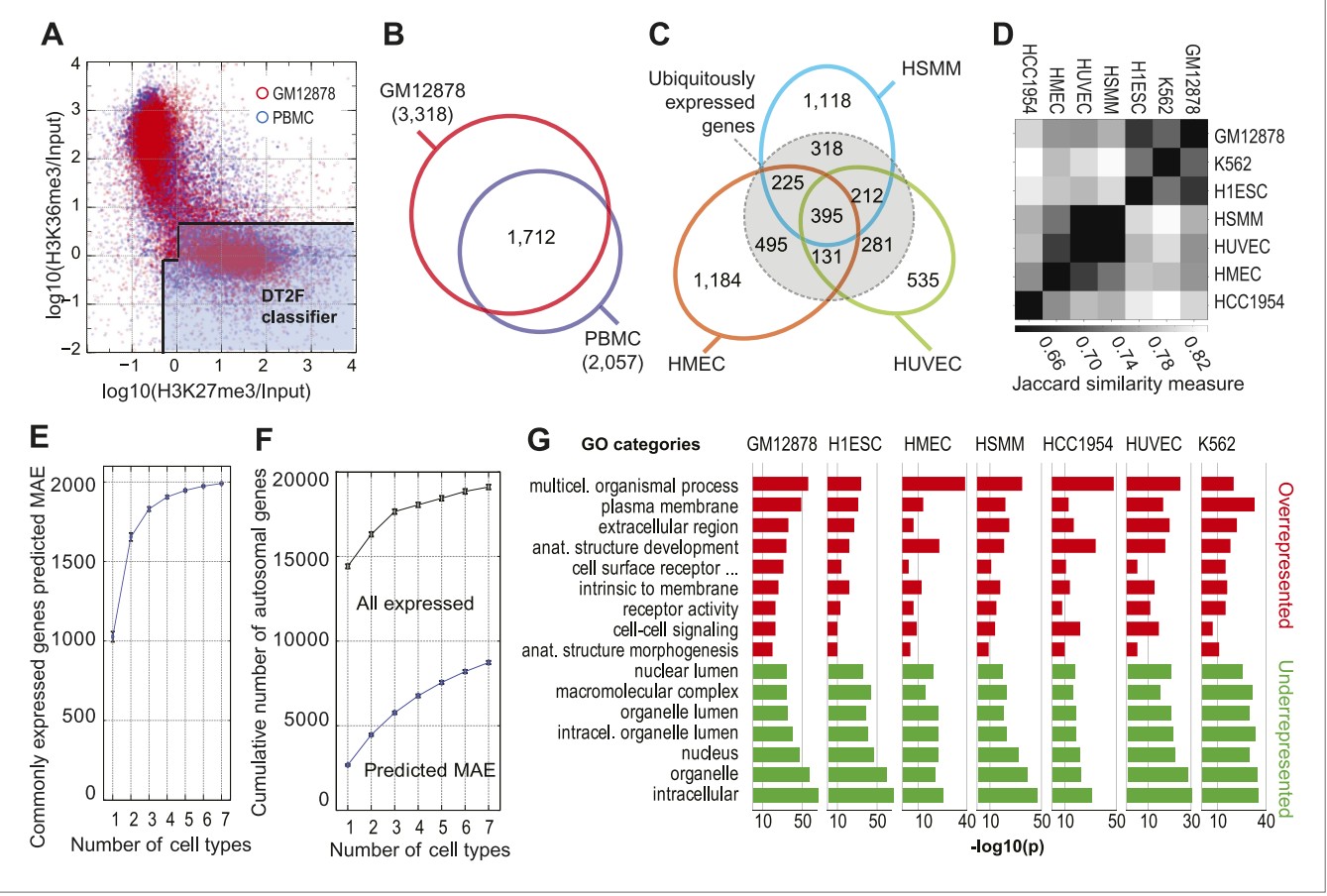

**Figure 5**. MAE chromatin signature in multiple cell types. (**A**) Comparison of H3K27me3 and H3K36me3 ChIP-Seq gene body signal distribution for the autosomal genes in GM12878 cells (red) and in primary peripheral blood monocytes (PBMC; blue). Silent genes (RPKM < 0.1) are excluded in either case. Both datasets were collected by ENCODE; PBMC data: GSE16368. Note that the dots are made more transparent than in *Figure 1* to make clear the overall shape of the distribution. (**B**) Overlap of predicted MAE genes in GM12878 and PBMC (silent genes are excluded). (**C**) Tissue-specific distribution of MAE genes. Overlap between predicted MAE genes in three cell types as labeled. Within dotted circle: genes expressed in all three lines (and MAE in at least one). Outside dotted circle: MAE genes showing cell type-specific expression (predicted MAE and expressed in that cell type, but silent in at least one of the two other cell lines). (**D**) Similarity of predicted MAE gene sets in seven cell types: GM12878—lymphoblasts, K562—acute myelocytic leukemia, H1ESC—human embryonic stem cells, HSMM—human skeletal muscle myocytes, HUVEC—human vascular epithelium, HMEC—human mammary epithelium, HCC1954—breast cancer. Similarity assessed according to the Jaccard similarity measure. In the heat map, darker gray corresponds to higher similarity. ChIP-Seq and RNA-Seq data sources: see Dataset S1 and S2 in Dryad (***Nag et al., 2013***). (**E**) Cumulative number of predicted MAE genes as a function of the number of cell lines assessed. Counted are only genes expressed in all analyzed cell lines. Order of addition of cell lines was sampled by permutation, shown are mean values ± standard deviation. (**F**) Cumulative number of all genes and predicted MAE genes as a function of the number of cell lines assessed. Counted are all genes with any evidence of expression in at least one cell type. (**G**) Gene Ontology (GO) analysis of genes predicted MAE in indicated cell types. Most over- and under-represented categories for GM12878 cells are also shown for other cell types (in each cell line, predicted MAE genes are compared to all expressed genes). Horizontal axis: −log10(p), after Benjamini–Hochberg correction. Gray lines correspond to −log10(p) values as noted.

The following figure supplements are available for figure 5:

**Figure supplement 1**. Overall distributions of the normalized H3K27me3 and H3K36me3 gene body signal in the analyzed cell types.

autosomal genes expressed in several cell types analyzed by ENCODE: vascular and mammary epithelial cells (HUVEC and HMEC), skeletal muscle (HSMM), embryonic stem cells (H1ESC), a leukemia line K562 and breast cancer line HCC1954, drawing on deposited data to analyze with our model: ChIP-Seq of H3K27me3, H3K36me3, Input control, and RNA-Seq data (*Figure 5C–D*). In the DT2F space, genes formed two major clusters in all assessed cell types (*Fig.5—figure supplement 1*). Although the precise positions of these clusters varied somewhat, the overall distribution remained consistent.

Importantly, the DT2F classifier trained on the GM12878 data consistently covered the lower cluster of genes in different assessed cell types.

In contrast to the similarity between PBMC and GM12878 data collected by different labs, cells of different types, showed pronounced differences in the set of genes classified by the DT2F as MAE, even though they were assessed in the same laboratory. This, along with the similarity in the general structure of the feature space (*Fig 5—figure supplement 1*) suggests that the biological properties of the cell type are captured by the model to a greater extent than other potential sources of variation and provides indirect support for the use of our method in other cell types. *Figure 5C* illustrates the prevalence of tissue-specific differences by comparing the predicted MAE genes in three cell lines. Genes predicted as MAE in only one of the three cell types are typically not expressed in all the three cell types. Strikingly, in each cell type between 1/3 and half of the predicted MAE genes showed cell type-specific expression. (In *Figure 5C*, compare 1184 predicted MAE genes expressed in HMEC but silent in one or both other cell lines, with 495 genes that are MAE only in HMEC but expressed also in HSMM and HUVEC cells, or with 395 genes that are expressed and predicted MAE in all three cell types.)

We interpret these observations to mean that the H3K36me3–H3K27me3 gene body signature is a general feature of monoallelic expression in multiple types of human cells, while a particular set of affected genes can be cell type-specific. It should be noted that the correspondence of the chromatin signature to monoallelic expression has only been systematically validated in clonal LCL lines (see *Figure 2* and *Figure 3*). Analysis in nonclonal cell populations (such as the ENCODE cell lines or isolated ex vivo cells) has been mostly restricted to single-cell techniques such as the fluorescent in situ hybridization approach which we have used in PBMCs (*Gimelbrant et al., 2007*). Until a more general study is completed, it remains formally possible that the chromatin signature does not reflect MAE in some or many cell types. The analysis in the rest of this section should be understood as being performed on genes that have a particular chromatin signature, which at least in one cell type strongly correlates with monoallelic expression.

The greater the biological similarity between cell types, the greater the overlap in MAE genes (*Figure 5D*). Overall, cell lines fell into two groups: GM12878, K562 and H1ESC had a larger percentage of transcribed genes with MAE chromatin signature (about 25%), while the rest had a lower percentage (about 15%). A large number of predicted MAE genes in ES cells suggest that the allelic choice may occur fairly early in development.

When assessing overlap between predicted MAE genes in different cell types, several patterns became evident. Only 61 genes were predicted as MAE in all of the seven analyzed cell types. Among commonly expressed genes (~10,000 genes with at least some evidence of expression in all analyzed cell lines), many are predicted as MAE only in some lines, implying tissue-specific MAE. Among ubiquitously expressed genes within this group, addition of more cell lines rapidly led to a plateau in the cumulative number of MAE genes, making up about 20% of the genes in this set (*Figure 5E*). In contrast, when all genes are considered (including cell type-specific ones), the number of genes expressed in at least one cell line and the cumulative number of predicted MAE genes keep rising, since a large fraction of the cell type-specific genes are predicted as MAE (*Figure 5F*). With seven cell lines, the total number of genes with evidence of expression in at least one cell line (RPKM>0.1) was 18,248, among which the total number of genes with the chromatin signature of MAE in at least one cell line was 8716 (48%). Extrapolation of this trend indicates that, with a large number of cell types, about half of all genes would be predicted to have MAE in one or more cell type. Qualitatively similar results remain if a higher expression threshold is applied, such as RPKM>1 (39% at seven cell lines).

Taken together, these analyses lead to two unexpected conclusions: (a) among widely expressed genes, about 20% have a chromatin pattern characteristic for MAE in at least one tissue; (b) taking genes with tissue-specific expression as a group, we estimate that between 39% and 48% have the MAE pattern.

## General properties of human MAE genes

The genes with MAE chromatin signature share some pronounced features as a group. Among predicted MAE genes, genes coding for cell surface proteins and those involved in multicellular developmental processes were heavily over-represented. Conversely, housekeeping genes and genes specific to intracellular organelles were highly under-represented. When the same gene ontology categories were assessed in other analyzed cells lines, they were similarly over- or under-represented at high significance

levels (*Figure 5G*). Since the overlap in actual genes is relatively modest, this suggests that in different cell types MAE affects different genes involved in the same type of activity.

A particularly important group consists of bivalent genes, which are also associated with overlapping active and inactive histone modifications. In embryonic stem cells, bivalent genes are transcriptionally silent, and their promoters carry overlapping peaks of H3K4me3 and H3K27me3 (*Mikkelsen et al., 2007*). These genes play a crucial role in determining cell fate during differentiation: upon reaching a point of lineage commitment, this 'poised' chromatin allows these genes to rapidly resolve into either an active or inactive state. To our surprise, we found that a remarkably large fraction of the known bivalent genes (>80%) acquire the MAE chromatin signature in at least one of the differentiated cell types, much higher even when compared to other genes that are also silent in ES cells (*Figure 6A*). Our observations indicate that upon activation, these master regulator genes preferentially resolve into a state with the characteristic MAE chromatin signature.

## Discussion

Using a systematic machine learning approach, we identified and then experimentally validated a specific and sensitive signature for MAE: a gene-body overlap between chromatin marks for active transcription (H3K36me3) and gene silencing (H3K27me3). Interestingly, the promoter signal carried very little additional information relative to the gene-body features (*Figure1—figure supplement 3*). This could be due to some combination of biological reasons with much lower noise associated with gene body features due to their much greater length. It should also be noted that we focused on promoters and gene bodies because they are unequivocally associated with particular genes. It might be informative to study the involvement of other regulatory elements, such as enhancers. Upon applying this approach to a variety of cell types, we estimate that up to 20% of ubiquitously expressed genes and more than one-third of tissue-specific genes showed the chromatin signature of MAE. Since detection of the MAE signature does not rely on SNPs or on sample clonality, we expect that this approach will make feasible the analysis of MAE in primary samples, including systematic comparisons of normal and diseased tissue.

Our findings fill three major gaps in our understanding of MAE. First, we have positively identified a molecular correlate of MAE with strong predictive power. We mapped MAE signature genome-wide in multiple cell types. Finally, we uncovered an unexpected relationship between MAE and known drivers of differentiation, bivalent genes, implying a functional role of MAE in this process.

The overlap of MAE and bivalent genes poses a potential mechanistic puzzle. In the promoters of bivalent genes, both active and inactive modifications are simultaneously present on both alleles (*Bernstein et al., 2006*). Moreover, direct evidence showed individual nucleosomes carrying H3K4me3 or H3K36me3 along with H3K27me3, albeit on the opposite histone H3 tails (*Voigt et al., 2012*). At the same time, asymmetric distribution of H3K27me3 and H3K36me3 within the gene body of MAE genes is consistent with a very simple asymmetric model: H3K27me3 is known to be associated with transcriptionally silenced chromatin, and H3K36me3 is associated with gene bodies of the actively transcribed genes (*Black et al., 2012*). This leads us to speculate that transition of the bivalent genes from transcriptionally silent state with 'poised' chromatin into an active state often happens independently for the two alleles. In that scenario, either allele can become stably inactive, with gene body chromatin enriched with H3K27me3, while the other becomes active and enriched with H3K36me3 (*Figure 6B*). Once the poised state has resolved into active or inactive state, it locks in that state; with some probability, both alleles become active, and the biallelic state is locked. While the molecular details of this process remain unclear, this model makes testable prediction that for bivalent genes, MAE is established during lineage commitment events, such as ES cell differentiation.

Our finding of a specific combination of molecular markers associated with autosomal MAE creates opportunities for a deeper understanding of mechanisms underlying this epigenetic phenomenon. The presence of H3K27me3 immediately implies that histone methyltransferase *EZH2*, and possibly other components of the PRC2 complex (*Margueron and Reinberg, 2011*), could be involved in MAE maintenance. Our experimental observations suggest that H3K27me3 and H3K36me3 marks are asymmetrically distributed between the alleles of genes with MAE. Our observations are consistent with the idea that a uniform molecular mechanism might be involved in MAE maintenance genome-wide.

Our study strongly implies that very similar sets of genes are subject to MAE in different individuals: the presence of the H3K36me3/H3K27me3 chromatin pattern in one individual corresponds to MAE in the same cell type in another individual. A parsimonious hypothesis is that the capacity for MAE is

**Figure 6**. MAE chromatin signature in bivalent genes. (**A**) Overrepresentation of bivalent genes among predicted MAE genes. Predicted: predicted MAE in at least one cell line and not silent (RPKM > 0.1); not predicted: not predicted MAE in any cell line. Groups of genes: (top) not bivalent, silent in hESC (RPKM < 0.1); (middle) not bivalent, not silent in hESCs; (bottom) reported bivalent in human ES cells. (**B**) A speculative model of MAE establishment in bivalent genes. Genes with bivalent/poised chromatin in promoters are silent in undifferentiated stem cells; two alleles have symmetric distribution of active and inactive histone marks. When such gene is activated upon reaching a point of cell fate determination, either one of the alleles (or both) can become transcriptionally active. The initial choice is stochastic, but it is stable in the clonal progeny. Asymmetric histone modifications in the gene body reflect activity of the alleles.

genetically controlled by regulatory DNA sequences, presumably in *cis* to the affected MAE genes. Consistent with this notion, significant inter-species conservation exists among genes subject to MAE in human and mouse (*Zwemer et al., 2012*).

Epigenetic regulation of monoallelic expression can profoundly affect the relationship between genotype and molecular phenotype (*Pereira et al., 2003*; *Zuo et al., 2007*). One intriguing possibility is raised by a recent observation that genomically identical clones can be dramatically different in terms of tumorigenic properties and drug resistance (*Kreso et al., 2012*). The knowledge of the chromatin signature of MAE will help in exploring the relationship between MAE and functional cellular states in normal development and disease.

## Materials and methods

### Automated classification of MAE

#### Data
The chromatin mark data for the ENCODE cell lines was obtained in wig format from the UCSC genome browser The chromatin mark data for HMEC and HCC1954 were obtained from Gene Expression Omnibus Series GSE29127 (*Hon et al., 2012*), and for PBMC were obtained from Gene Expression Omnibus Series GSE16368 (*Bernstein et al., 2010*). The expression data for all ENCODE cell lines, including GM12878, were obtained from UCSC ENCODE DCC, Gene Expression Omnibus Series GSM958730 (GM12878), GSM958737 (H1hESC), GSM958744 (HSMM), GSM958738 (HUVEC) and GSM958731 (K562).

#### Feature generation
To generate features, continuous signal from ChIP-Seq with a given antibody was integrated over proximal promoter (2.5 kb upstream of the transcription start site), or the whole gene body, and normalized to the signal from input (whole cell extract; WCE). The data were processed using in-house *perl* and *awk* scripts. The raw coverage counts of chromatin marks, and input on the promoter and on the gene body were calculated by summing the signal over the respective area of interest. The resulting raw amount of each mark was normalized to the input as the log2 ratio of mark to input.

#### Classifier training, evaluation and prediction
Genes were split into two sets: training/development and prediction, where training/development set contained genes with known MAE or biallelic status, and the prediction set contained all genes with unknown status. Additional quantile normalization was performed for HMEC and PBMC datasets,

using *matlab*. Our training/development set consisted of 270 high confidence MAE genes and 1068 high confidence BAE genes (with at least four clones showing biallelic expression) (*Gimelbrant et al., 2007*). We used all genes for which the MAE or biallelic status was ascertained with high confidence to gain maximum information from all available knowledge, including negative examples. Note that not all genes had data for every feature tested in various combinations. For example, after removing genes with no data, there were 266 MAE and 1046 BAE genes with H3K27me3 and H3K36me3, and 706 BAE and 171 MAE genes in the complete feature set.

We used *Weka 3.7.3* (*Hall et al., 2009*) to train and evaluate classification methods and perform automated feature selection. Initially, the full set of features was provided and greedy hill-climbing approach augmented with a backtracking facility (BestFirst method in Weka: Select Attributes) was used for feature search. The chosen feature subsets were evaluated by considering the individual predictive ability of each feature along with the degree of redundancy between them (CfsSubsetEval method in Weka: Select Attributes). In addition, the features were evaluated individually using information gain with respect to the class (InfoGainAttributeEval in Weka: Select Attributes) and ranked accordingly. The top features were added one by one until no significant gain in performance was observed. In practice, the top two features H3K36me3 and H3K27me3 on the gene body were sufficient to achieve performance on par with the full feature set.

For evaluation purposes, we used 10-fold cross-validation, a procedure by which the training set is divided into 10 random subsets 10 different times, and each time the classifier is trained on nine subsets and tested on the remaining subset. The F-measure (harmonic mean of precision and recall) was taken as a measure of classifier performance. We trained the best classifiers on the full training/development set and saved the models in *Weka* model files.

For prediction purposes, we ran *Weka* in prediction mode with the saved model files on the prediction set, and obtained the results as CVS files which were processed in Excel (Data reported in Dataset S1 in Dryad, *Nag et al., 2013*).

The best and most parsimonious classifier was an alternating decision tree on H3K327me3 and H3K36me3 gene body signal, with default parameters (number of boosting iterations 10, searchpath = 'expand all paths'). An alternating decision tree consists of decision nodes and prediction nodes. 'Decision nodes' specify a predicate condition (e.g., log10H3K27me3 > 1). 'Prediction nodes' contain a single number. ADTrees always have prediction nodes as both root and leaves. An instance (gene) is classified by the ADTree by summing any prediction nodes that are traversed while following all paths for which all decision nodes are true. Varying the boosting iterations between 10 and 20 did not substantially alter the results and produced a decline outside of the range.

## Expression analysis

Alignment of RNA-seq data was done using Bowtie2 (*Langmead and Salzberg, 2012*), using paired end alignment, seed length = 28, and max seed mismatches = 2. RPKM was calculated using *cufflinks v. 2*, with a gtf file for *Homo sapiens*, UCSC, hg19 obtained from http://cufflinks.cbcb.umd.edu/igenomes.html. For comparison, *cufflinks* was run using the multiple aligned files option simultaneously on all aligned files (bam format) of the ENCODE cell lines. The HMEC and HCC1954 were subsequently run together using the same option separately from the rest and quantile normalization to GM12878 reference was performed on the RPKM values.

## Extraction of allelic counts

In-house analysis pipeline in *perl* and *awk* was used to generate an SNP-masked reference and to obtain mapped read counts for each SNP. The reference was derived from the hg19 genome by removing non-transcribed regions using gtf annotation, and masking SNP loci. Long-distance bias effects (*McDaniell et al., 2010*) were also removed using in silico sequencing simulation. The reads covering SNP loci were tallied into maternal and paternal 'hits'.

In-house *Matlab* analysis pipeline was used to calculate binomial p-value with FDR-correction and perform equivalence testing for each gene. Allelic bias was statistically identified from the resulting SNP allelic counts. As a first step, counts for multiple SNPs spanning the same gene were tallied, since SNPs were never less than one read length apart and each hit was therefore considered the result of an independent Bernoulli trial of a random variable representing gene bias, with complete maternal bias corresponding to parameter value 0, lack of bias to value 0.5, and complete paternal bias to value 1.

In processing AST-seq data, which had very high coverage per SNP (generally >$10^3$), we used equivalence testing and rejection of equivalence as the test for BAE and MAE with equivalence boundaries of 2:1 bias (corresponding to the maximum likelihood estimate parameter value of less than 0.33 or more than 0.67). Due to the much lower coverage in the whole genome RNA-seq experiment, we used FDR-corrected binomial testing of the tallied counts for each of the heterozygous genes for which hits were obtained, along with a threshold of 2:1 bias (corresponding to the maximum likelihood estimate parameter value of less than 0.33 or more than 0.67) to filter out statistically significant but biologically weak bias. To call unbiased (biallelic) genes in each clone, we used equivalence testing, with equivalence boundaries corresponding to the above 2:1 bias. While the use of 2:1 bias as threshold is conventional (*Zwemer et al., 2012*; *Li et al., 2012a*), the results remained qualitatively robust with other thresholds (e.g., 3:1) used. This was due to the fact that a disproportionate fraction of MAE genes showed extreme biases.

In the analysis of data from the multiplex capture of common coding SNPs using padlock probes (see below), we restricted the analysis to SNPs with 100 reads or more.

### Gene set enrichment analysis
Gene set enrichment analysis was performed using the GeneTrail online tool with methodology described in *Keller et al. (2007)*.

## Cell lines
### Cell culture
GM12878 cell line was obtained from Coriell Cell Repositories. GM13130 polyclonal cell line and GM13130 clones were described before (*Gimelbrant et al., 2007*).

### Single cell cloning
Fluorescence Activated Cell Sorting (FACS) of single live GM12878 cells was performed following Propidium Iodide (PI) staining (5 µg/M cells). We used two conditions for culturing sorted single cells: (a) 50% conditioned media and (b) Mitomycin C (10 and 50 µg/ml) inactivated feeder cells. To control for possible escape of feeder cells from cell-cycle arrest, we used as feeders lymphoblasts from a different individual (GM13130) and genotypes all clones to ensure their identity. Clone expansion was evaluated 21–24 days post sorting.

### Establishing independence of clones
To identify clones that had unique rearrangement in Igκ or Igλ locus, we performed degenerate RT-PCR as previously described (*Wardemann et al., 2003*).

### Assessment of genome integrity
To assess for gross genomic abnormalities, unique clones were subjected to metaphase spread analysis according to standard protocol (*Deng et al., 2003*). To assess for smaller-scale changes, we performed SNP6.0 genotyping ana analysis from these clones. Briefly, DNA was extracted from the clones using Qiagen blood mini kit and was subjected to SNP6.0 genotyping according to standard protocol at Vanderbilt Microarray Shared Resource (http://array.mc.vanderbilt.edu/). The data were analyzed using standard *Birdseed* pipeline (*Korn et al., 2008*).

## Deep sequencing
### Strand specific genome wide RNA sequencing
Strand-specific RNA-sequencing library was prepared according to *Parkhomchuk et al. (2009)*. Briefly, total RNA was extracted using Trizol reagent (Life Technologies, Carlsbad, CA) using standard protocol. Total RNA was subjected to polyA selection using Poly(A)Purist MAG Kit (Life Technologies). This RNA was DNase treated followed by first strand synthesis in the presence of Actinomycin D and second strand synthesis in the presence of dUTP instead of dTTP. Sheared cDNA (Covaris Inc, Woburn, MA) was end-repaired and subsequently adenylated. Adapters were ligated and ran on gel to cut and elute the pieces of desired size. The eluted DNA was subjected to a treatment by USER enzyme (NEB, Ipswich, MA) to remove the second strand. The resulting mixture was PCR amplified (using primers recommended by Illumina but synthesized by IDT, Coralville, IA) using Phusion High-Fidelity DNA Polymerase (NEB) for 15 cycles followed by agarose gel purification of the band. The library was sequenced using Illumina HiSeq 2000 platform.

## AST-Seq

ach of the gene-specific primers flanking the targeted SNP had one of two universal tails (F tail: 5′GCG TAC CAC GTG TCG ACT or R tail: 5′GAC GGG CGT ACT AGC GTA). Second round of PCR used these universal tails to introduce unique combinations of 6-nucleotide 'barcode' sequences. Genomic DNA (gDNA) was isolated using Qiagen Blood kit. RNA was isolated using Trizol reagent (Life Technologies) using standard protocol. DNase-treated (Turbo, Ambion, Life Technologies) RNA was used to prepare cDNA using SuperScriptIII (Life Technologies). PCR was performed using Klear Taq (KBioscience, UK) (95°C, 15 min enzyme activation followed by 35 cycles of 95°C, 30 s denaturation; 62°C, 30 s annealing; 68°C, 90 s extension; and a final extension of 72°C for 5 min). After two rounds of PCR, barcoded amplicons were mixed in equal amount for preparation of the single library. The mixed amplicons were size-selected on agarose gel, end repaired, adenylated, and finally adapters were ligated. The resulting DNA fragments of desired size range were purified on agarose gel, and pre-amplified for 15 cycles. The library was mixed with high-complexity libraries (20–30%) and subjected to paired-end or single-end sequencing for 100 cycles on Illumina HiSeq 2000. Primer sequences are listed in Dataset S3 in Dryad (*Nag et al., 2013*).

## Chromatin immunoprecipitation

ChIP was performed as described (*Bernstein et al., 2006*; *Mikkelsen et al., 2007*). Chromatin shearing was performed using Covaris system. The antibodies used were ABE44 (Millipore, Billerica, MA) for H3K27me3 and AB9050 (Abcam, Cambridge, MA and UK) for H3K36me3.

## Multiplex capture of common coding SNPs

### Experimental method

To capture the targeted coding SNPs, mixture containing Padlock probes targeting 36,456 common coding SNPs and 200 ng of genomic DNA (or cDNA, ChIPed-DNA) was prepared in 1X Ampligase Buffer (Epicentre, Madison, WI) (*Zhang et al., 2009*). The mixture was denatured at 95°C, gradually cooled and then hybridized at 60°C for 24 hr. The product was circularized after adding AmpliTaq Stoffel (Life Technologies), and Ampligase (Epicentre), in the presence of dNTPs, and incubating at 60°C for 18 hr. Following this, exonuclease I (USB, Affymetrix, Santa Clara, CA) and exonuclease III (USB) were added to cleave the linear DNA by incubating at 37°C for 2 hr and followed by heat inactivation at 90°C. The circularization product was amplified using primer AmpF6.3Sol: AATGATACGGCGACCACCGACACTCTCAGATGTTATCGAGGTCCGAC, AmpR6.3Ind in Kapa SYBR FAST qPCR master mix (Kapa Biosystems, Wilmington, MA) using the following program: 95°C for 2 min, and then six cycles at 98°C for 20 s, 58°C for 20 s, 72°C for 20 s followed by 20 cycles at 98°C for 10 s and 72°C for 20 s. PCR reaction was terminated before the real time PCR curve reached plateau to avoid over-amplification. The amplicons were purified using Qiaquick PCR purification kit (Qiagen, Netherlands), and the products in the expected size range (180 bp) were selected using polyacrylamide (PAGE) gel. The libraries were sequenced on an Illumina Genome Analyzer IIx.

### Read mapping and extraction of allelic counts

Sequencing reads generated were trimmed 9 bp from both ends, mapped to the human genome (hg19) with *bwa* (*Li and Durbin, 2009*), then quality-recalibrated and locally re-aligned with GATK. Heterozygous SNPs were identified from genomic DNA using GATK, and the allelic counts at these sites were extracted from the *bam* files using Samtools (*Li et al., 2009*).

### Method note 1: relationship of RPKM value and number of transcripts per cell

We estimate that in lymphoblastoid cells RPKM of 0.1 corresponds to about 0.5–1 transcript per cell. We based our gene expression cutoff of 0.1 RPKM for expressed genes on several considerations: our experimental cell lines are lymphoblastoid, which are smaller than average sized cells. While 100 ng of RNA of cells used by (*Mortazavi et al., 2008*) correspond to approximately $10^3$ cells, we require between $1–3 \times 10^4$ cells, to obtain the same amount of RNA. Hence, if (*Mortazavi et al., 2008*) calculate that RPKM = 1 corresponds to approximately one transcript per cell based on their spike-in data, we adjust the empirical curve by a factor of 10 to obtain 0.1 RPKM to stand for approximately one transcript per cell. In addition, we note that while only 9% of genes with RPKM of 0 in the data for the polyclonal line are detected in our clonal lines, 50% of the genes with 0.1 RPKM are detected (*Rozowsky et al., 2011*). This means that we have direct evidence that at least half of the genes at that level are represented at

one transcript per every two (clonal) cells. Given that detection is not perfect, we believe this to be a very conservative estimate. Last but not least, 0.1 RPKM as a detection threshold for whole transcriptome analysis is a widely accepted method (*Lundberg et al., 2010*; *Uhlen et al., 2010*).

## Method note 2: choice of genes for AST-seq validation

The genes were chosen based on several criteria. First we created a list of genes that had heterozygous SNPs in the primary transcript in both the GM12878 and GM13130 lines. We then looked at dataset generated by the Wold and Gerstein labs on the GM12878 individual to create a subset of genes that were expressed in this cell type. We had very few genes in the predicted MAE table that had heterozygous SNPs in the primary transcript in both the cell lines and were expressed at sufficient level. Many MAE genes chosen also had multiple SNPs per transcript. The BAE genes to be tested were then chosen randomly from the resulting table. These were the genes that were NOT predicted to be MAE.

## Acknowledgements

We thank A Bortvin, M Kuroda, R Medzhitov, F Winston, and members of the Gimelbrant lab for critical comments on the manuscript, A Alekseenko, B Bernstein, and A Regev for helpful technical suggestions, and SA Gimelbrant, A Landry, JB Lazaro and R Issner for technical help.

## Additional information

### Funding

| Funder | Grant reference number | Author |
|---|---|---|
| Claudia Adams Barr Foundation | | Alexander A Gimelbrant |
| Susan F Smith Center for Women's Cancers | | Alexander A Gimelbrant |
| National Institutes of Health | R01GM097253 | Kun Zhang |
| Pew Scholars Program | | Alexander A Gimelbrant |

The funders had no role in study design, data collection and interpretation, or the decision to submit the work for publication.

### Author contributions

AN, VS, Conception and design, Acquisition of data, Analysis and interpretation of data, Drafting or revising the article; H-LF, AM, Acquisition of data, Analysis and interpretation of data; G-CY, Critical reading of the article, Analysis and interpretation of data; KZ, Acquisition of data, Contributed unpublished essential data or reagents; AAG, Conception and design, Analysis and interpretation of data, Drafting or revising the article

## Additional files

### Major dataset

The following datasets were generated:

| Author(s) | Year | Dataset title | Dataset ID and/or URL | Database, license, and accessibility information |
|---|---|---|---|---|
| Nag A, Savova V, Fung H, Miron A, Yuan G, Zhang K, Gimelbrant AA | 2013 | Data from: Chromatin signature of monoallelic expression | 10.5061/dryad.1775k | Publicly available at Dryad (http://datadryad.org/). Dataset S1, MAE predictions for cell lines; Dataset S2, Result of whole RNA-seq on clones of GM12878 lymphoblastoid clones DF1 and DF2; Dataset S3, AST-seq data; Dataset S4, Results of the experiment with padlock probe. |

| Author(s) | Year | Dataset title | Dataset ID and/or URL | Database, license, and accessibility information |
|---|---|---|---|---|
| Nag A, Savova V, Gimelbrant AA | 2013 | Whole RNA-seq on clones of GM12878 lymphoblastoid clones DF1 and DF2 | GSE52090; http://www.ncbi.nlm.nih.gov/geo/query/acc.cgi?acc=GSE52090 | Publicly available at GEO (http://www.ncbi.nlm.nih.gov/geo/) |
| Fung H-L, Zhang K, Nag A, Savova V, Gimelbrant AA | 2013 | Data for allelic bias analysis in gDNA, cDNA and ChIP with H3K27me3 and H3K36me3 antibodies with multiplexed padlock probe approach | GSE53628; http://www.ncbi.nlm.nih.gov/geo/query/acc.cgi?acc=GSE53628 | Publicly available at GEO (http://www.ncbi.nlm.nih.gov/geo/) |

The following previously published datasets were used:

| Author(s) | Year | Dataset title | Dataset ID and/or URL | Database, license, and accessibility information |
|---|---|---|---|---|
| Dunham I, Kundaje A, Aldred SF, Collins PJ, Davis CA, Doyle F, et al. | 2012 | Data from: An integrated encyclopedia of DNA elements in the human genome | http://genome.ucsc.edu/ENCODE; http://www.ncbi.nlm.nih.gov/geo/info/ENCODE.html | Publicly available at GEO (http://www.ncbi.nlm.nih.gov/geo/). |
| Bernstein BE, Stamatoyannopoulos JA, Costello JF, Ren B, Milosavljevic A, Meissner A, et al. | 2012 | Data from: The NIH Roadmap Epigenomics Mapping Consortium | GSE16368; http://www.ncbi.nlm.nih.gov/geo/query/acc.cgi?acc=GSE16368 | Publicly available at GEO (http://www.ncbi.nlm.nih.gov/geo/). |
| Hon GC, Hawkins RD, Caballero OL, Lo C, et al. | 2012 | Global DNA hypomethylation coupled to repressive chromatin domain formation and gene silencing in breast cancer | GSE29127; http://www.ncbi.nlm.nih.gov/geo/query/acc.cgi?acc=GSE29127 | Publicly available at GEO (http://www.ncbi.nlm.nih.gov/geo/). |

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
