## [Decision Letter]

Thank you for sending your work entitled “Chromatin signature of widespread monoallelic expression” for consideration at *eLife*. Your article has been favorably evaluated by a Senior editor and 3 reviewers, one of whom is a member of our Board of Reviewing Editors.

The Reviewing editor and the other reviewers discussed their comments and the Reviewing editor has assembled the following list of areas of clarification and editorial comments that should be addressed.

1) The authors mentioned that they have explored different features except H3K27me3 and H3K36me3 to classified MAE and BAE genes, but do not describe them. However, it is important to see how other features affect the performance of classification.

2) The rationale of selecting several thresholds used in the analyses is not discussed nor is the variability of the results described if other higher or lower thresholds are used. This question can be applied to the 2:1 ratio used for RNAseq data to distinguish MAE from BAE genes and the thresholds corresponding to the log H3K36me3/input vs log H3K27me3/input inside of Figure 1.

3) The predictability of the signature studied should be addressed by either providing information concerning results obtained from other cell types or make more explicit that since this approach was carried out with only a very limited biological context the general applicability is currently uncertain.

4) The authors describe the chromatin signature of gene bodies. It is not clear why they focused on gene bodies with no mention of promoter versus gene body status, or of other possible regulatory elements.

5) Why did the authors not choose similar numbers of BAE genes (1068) as MAE (270) to train the model?

6) Why are only 27% of X-linked genes showed positive evidence of clone-specific allelic bias? Surely this should be higher if this number is of the total expressed genes? How many of the X-linked genes looked at are known escapees?

7) For Figure 1 and Figure 4 what percentage of the previously identified monoallelically expressed genes have both chromatin signatures and the number of the genes assayed needs to be included in Figure 4.

8) Although the authors attempt to deal with this in the Discussion, it is difficult to understand molecularly how bivalent genes (which are stated to be overrepresented in the predicted monoallelically expressed genes) resolve into a monoallelically expressed gene predicted chromatin signature.

9) The subsection title “Human MAE subgenome” is obscure and should be clarified.

---

## [Author Response]

*1) The authors mentioned that they have explored different features except H3K27me3 and H3K36me3 to classified MAE and BAE genes, but do not describe them. However, it is important to see how other features affect the performance of classification*.

We have now added Figure 1—figure supplement 3 with multiple panels showing performance of other gene body features, as well as promoter features, with the training set genes. The main text has been adjusted accordingly with three new paragraphs starting:

“We focused on the eight marks that were investigated in a broad variety of cell types: H3K27me3 (histone H3 Lys-27 trimethylation), H3K36me3, H3K4me2, H4K20me3, H3K27ac (histone H3 Lys-27 acetylation), H3K4me1, H3K4me3, H3K9ac (Figure 1)…”

This also addresses one part of point 4.

*2) The rationale of selecting several thresholds used in the analyses is not discussed nor is the variability of the results described if other higher or lower thresholds are used. This question can be applied to the 2:1 ratio used for RNAseq data to distinguish MAE from BAE genes and the thresholds corresponding to the log H3K36me3/input vs log H3K27me3/input inside of*
Figure 1.

We would like to clarify that the minimum 2:1 ratio of allele-specific counts in RNA-Seq data was used as one necessary but not sufficient criterion of allelic bias (Figure 2 summarizes additional statistical tests we required). Moreover, presence of allelic expression bias, no matter how strong, was in and of itself insufficient to call a gene MAE. In order to distinguish MAE-based bias from *cis*-regulatory or imprinting-based bias, we also required positive identification of equal or preferential expression of the other allele in other clonal cell populations (see Figure 3 for examples).

We have edited the text in the Materials and methods and Results sections to reflect this.

*3) The predictability of the signature studied should be addressed by either providing information concerning results obtained from other cell types or make more explicit that since this approach was carried out with only a very limited biological context the general applicability is currently uncertain*.

We clarified the text as follows:

“It should be noted that the correspondence of the chromatin signature to monoallelic expression has only been systematically validated in clonal LCL lines (see Figure 2 and Figure 3). Analysis in nonclonal cell populations (such as the ENCODE cell lines or isolated *ex vivo* cells) has been mostly restricted to single-cell techniques such as the fluorescent in situ hybridization approach which we have used in PBMCs (17). Until a more general study is completed, it remains formally possible that the chromatin signature does not reflect MAE in some or many cell types. Thus the analysis in the rest of this section should be understood as being performed on genes that have a particular chromatin signature, which at least in one cell type strongly correlates with monoallelic expression.”

*4) The authors describe the chromatin signature of gene bodies. It is not clear why they focused on gene bodies with no mention of promoter versus gene body status, or of other possible regulatory elements*.

We discuss analysis of promoter features in response to point 1. We also added the following to the Discussion:

“Interestingly, the promoter signal carried very little additional information relative to the gene-body features (see Figure 1—figure supplement 3). This could be due to some combination of biological reasons and higher signal/noise ratio associated with gene body features due to their much greater length. It should also be noted that we assessed promoters and gene bodies because they are unequivocally associated with particular genes. In the future, it might be informative to study the role of other regulatory elements, such as enhancers.”

*5) Why did the authors not choose similar numbers of BAE genes (1068) as MAE (270) to train the model*?

To clarify this point, we added the following into the “Classifier training, evaluation and prediction” subsection of the Materials and methods:

“We used all genes for which the MAE or biallelic status was ascertained with high confidence to gain maximum information from all available knowledge, including negative examples.”

*6) Why are only 27% of X-linked genes showed positive evidence of clone-specific allelic bias? Surely this should be higher if this number is of the total expressed genes? How many of the X-linked genes looked at are known escapees*?

We showed that fraction with respect to all X-linked genes with any coverage, rather than those that had sufficient coverage on the SNPs. That was confusing and we changed the description to include only genes that were sufficiently covered. We adjusted the description accordingly.

*7) For*
Figure 1
*and*
Figure 4
*what percentage of the previously identified monoallelically expressed genes have both chromatin signatures and the number of the genes assayed needs to be included in*
Figure 4.

We added the following text to the caption of Figure 1: “Of 270 high-confidence MAE genes, 268 had data for both H3K27me3 and H3K36me3. Of these, 204 (76%) are within predicted MAE region.”

And we have added the following to the caption of Figure 4: “Analysis summarized in this figure is based on 482 SNPs within 458 genes.”

*8) Although the authors attempt to deal with this in the Discussion, it is difficult to understand molecularly how bivalent genes (which are stated to be overrepresented in the predicted monoallelically expressed genes) resolve into a monoallelically expressed gene predicted chromatin signature*.

We rewrote the corresponding paragraph to reflect this.

*9) The subsection title “Human MAE subgenome” is obscure and should be clarified*.

We changed the section subtitle to read: “General properties of human MAE genes”